Increased glycoprotein hormone yield in stably transfected CHO cells using human serum albumin signal peptide for beta-chains

http://orcid.org/0000-0002-8950-6061 Sinegubova Maria V. mvsineg@gmail.com
http://orcid.org/0000-0002-9230-3433 Kolesov Denis E.
http://orcid.org/0000-0003-0064-4458 Vorobiev Ivan I.
http://orcid.org/0000-0001-9483-2892 Orlova Nadezhda A.
Laboratory of Mammalian Cell Bioengineering, Institute of Bioengineering, Research Center of Biotechnology of the Russian Academy of Sciences , Moscow , Russia
Banerjee Priyanka
Electronic publication date: 2025 Feb 14
Publication date: 2025
Volume: 13
Electronic Location ID: e18908
Received 2024 Aug 16; Accepted 2025 Jan 6
Copyright: © 2025 Sinegubova et al.
Copyright year: 2025
Copyright holder: Sinegubova et al.
License: This is an open access article distributed under the terms of the Creative Commons Attribution License, which permits unrestricted use, distribution, reproduction and adaptation in any medium and for any purpose provided that it is properly attributed. For attribution, the original author(s), title, publication source (PeerJ) and either DOI or URL of the article must be cited.
License URL: https://creativecommons.org/licenses/by/4.0/

Keywords: Glycoprotein hormones, Gonadotropins, Human follicle stimulating hormone (hFSH), Human luteinizing hormone (hLH), Human chorionic gonadotropin (hCG), Human thyroid stimulating hormone (hTSH), Signal peptide, Recombinant protein production, CHO cell culture

Funding: Ministry of Science and Higher Education of the Russian Federation The work was performed with funding from the Ministry of Science and Higher Education of the Russian Federation. The funders had no role in study design, data collection and analysis, decision to publish, or preparation of the manuscript.

==============================
Heterologous signal peptides enable increasing titers of recombinant proteins in mammalian cell culture. Four human heterodimeric glycoprotein hormones (follicle-stimulating hormone, FSH; luteinizing hormone, LH; chorionic gonadotropin, CG; and thyroid-stimulating hormone, TSH) were expressed in stably transfected CHO cells when varying signal peptides of their β-subunits. The signal peptide of human serum albumin proved to be the most effective for the glycoprotein hormone family. The cell specific productivity was increased for LH (2.5 pg/cell, 4-fold increase), TSH (1.6 pg/cell, 13-fold increase), and CG (1.0 pg/cell, 60%-increase). According to the Western blotting and quantitative PCR data, the productivity increase is associated with an increase in the efficiency of translation and translocation of β-subunits of hormones in the endoplasmic reticulum due to their coupling with the heterologous signal peptides.

Introduction

A glycoprotein hormone family in humans includes three pituitary hormones—human thyroid-stimulating hormone (hTSH), human follicle-stimulating hormone (hFSH), human luteinizing hormone (hLH)—and placental hormone human chorionic gonadotropin (hCG). In the human body these hormones regulate the activities of the gonads (FSH, LH, CG, therefore referred to as gonadotropins) and thyroid gland (TSH). As drugs, FSH, LH, and CG are used in assisted reproductive technologies (Lunenfeld et al., 2019), TSH mainly in diagnostic testing for thyroid cancer (Pacini & Castagna, 2008). Nowadays these hormones are widely manufactured as recombinant biotherapeutic proteins, mostly in Chinese hamster ovary (CHO) cell culture (Leão & Esteves, 2014).

All four members are structurally related heterodimers consisting of two noncovalently associated subunits (α and β) with a high degree of glycosylation and cause physiological effects only as a αβ-heterodimer (Cahoreau, Klett & Combarnous, 2015). The α-subunit is common for the entire glycoprotein hormone family, and the β is unique for each hormone. Both subunits are translated as pre-proteins paired with an appropriate signal peptide (SP), a short N-terminal amino acid sequence facilitating co-translational translocation of the translated polypeptide into the endoplasmic reticulum (ER) (Liaci & Förster, 2021). The conversion of pre-α and pre-β subunits containing signal sequences into their mature forms involves two events: SP cleavage and glycosylation. Cleavage of the SP occurs co-translationally (Jackson & Blobel, 1980), while glycosylation occurs both co- and post-translationally (Weintraub et al., 1980; Ruddon & Bedows, 1997). Folding and assembly of the αβ subunits occurs within the ER lumen (Magner & Weintraub, 1982; Hoshina & Boime, 1982).

A complicated and cost-intensive bioprocessing of recombinant hormones in the CHO cell culture demands new upstream technologies to enhance products titers. Cell specific productivity may be limited on the level of: transcription, translation, processing, secretion, and stability of the protein in the culture medium. One of the most widely used approaches is enhancing transcription, but it has been shown that the mRNA levels don’t always correlate with the expression of the protein of interest (Barnes, Bentley & Dickson, 2004; Brion, Lutz & Albert, 2020). Another limiting step for secreted proteins biosynthesis is translocation of the translated polypeptide into the ER, which can be facilitated by replacing a native SP with an alternative one (Zhang, Leng & Mixson, 2005; Tan, Ho & Ding, 2002). SPs from different species are functionally interchangeable, and for CHO cells it has been shown that a heterologous SP may be more effective than a native one (Zhang, Leng & Mixson, 2005; Knappskog et al., 2007; Tan, Ho & Ding, 2002). Kober, Zehe & Bode (2013) observed enhanced expression levels for the recombinant proteins (an antibody and a fusion protein) secreted by CHO K1 cells when using signal sequences of human proteins. Synthetic SPs are also being currently designed (Yu et al., 2022; Park et al., 2022).

Currently existing bioinformatics services (SignalP 6.0) allow to predict whether a SP-protein combination can be cleaved accurately (Teufel et al., 2022). However, these algorithms do not predict the translation rate of the corresponding mRNAs and cannot distinguish effective SPs from ineffective ones. The development of a regression model for the secretion efficiency prediction based on the sequence of a SP and the first 50 amino acids of the target protein is reported in O’Neill et al. (2023). For a single-chain fragment variable fusion protein, the authors managed to select two SPs from a sample of 40,000 amino acid sequences that led to a 2-fold titer increase. At the same time, the authors report that the acceptable predictive efficiency of the model (R2 = 0.65) was achieved only for single-chain proteins, but not for monoclonal antibodies, due to the complexity of balanced synthesis of light and heavy chains (O’Neill et al., 2023).

There are several ways for the expression of two independent heterologous genes-of-interest: (a) two different plasmids with different selection markers; (b) one plasmid with tandem arrangement of two open reading frames (ORF) with two promoters and a common selection marker; (c) one plasmid in which a pair of GOIs is expressed from one promoter as part of polycistronic mRNA; (d) biosynthesis of two proteins encoded in one cistron followed by 2A-peptide-mediated cleavage (Davies et al., 2011; Chng et al., 2015). In addition, several variants of covalently cross-linked single-chain glycoprotein hormones have been obtained (Nguyen, Klett & Combarnous, 2023; Byambaragchaa et al., 2024). Not all variants of the cross-linked heterodimers demonstrate biological activity due to the importance of the relative position of the α- and β-chains. In natural glycoprotein hormones, the β-subunit seatbelt formed by cysteine residues embraces the α-subunit and ensures the αβ-heterodimer stability in physiological conditions (Nguyen, Klett & Combarnous, 2023). Moreover, these fusion glycoproteins are potentially immunogenic (Nguyen, Klett & Combarnous, 2023). Probably in the near future such glycoproteins will find application in veterinary medicine, for example, in 2020 a patent for single-chain CG in horses was registered (Pérez Sáez & Bussman, 2019).

Previously, we have found that when FSH subunit genes were expressed in CHO cells as part of the tricistronic plasmid, the cells predominantly secreted the free α-subunit rather than the heterodimeric hormone (Orlova et al., 2019). However, we didn’t observe large amounts of free α-subunit in the culture medium after the cells were re-transfected with a plasmid encoding the FSH β-subunit gene and an additional selection marker (Orlova et al., 2019). Such a method of balancing the subunits expression levels requires a long time for cell line development and, apparently, does not allow selecting the most productive cell clones due to the independent distribution of hormone subunit gene expression levels in individual cells.

The β-chains of CG and LH are almost identical: they share 85% sequence homology, CG has an additional C-terminal domain of 30 amino acids with four additional sites of O-glycosylation (Choi & Smitz, 2014). The FSH β-chain and the TSH β-chain have relatively little mutual homology. We assumed that optimal SPs might prove to be effective for the β-chains of the hormones which natural biosynthesis level in the body is very low (LH, TSH), and are less effective for FSH and CG. In our previous study, we investigated the influence of heterologous SPs in the β-chains of glycoprotein hormones on the biosynthesis in transiently transfected CHO S cells (Sinegubova et al., 2024). For the β-chains of all four hormones, we tested native SPs of their β-chains (NSPs in all cases) and heterologous SPs, taken from human serum albumin (HSA), human azurocidin (Azu), and a native SP of the common for all glycoprotein hormones α-chain (aSP). When replacing the natural SPs of the β-chains with the heterologous SP of HSA (HSA-SP), in the transiently transfected CHO cells we detected the significant increase in the specific productivity, 2–2.5 times for LH, CG, TSH hormones, and virtually no change for FSH. No significant increase in specific productivity was observed for Azu-SP and αSP. However, the levels of target protein expression are largely different for transiently transfected cells and stably transfected gene-amplified pools, usually by 2–3 orders of magnitude, so the transient expression data can’t predict the specific productivity of stably transfected cells due to, for example, change in the rate-limiting step of the protein expression pathway (Reinhart et al., 2014). It is already known that the performance of SP in transiently transfected cells and in stable cell lines might differ (Kalwy, Rance & Young, 2006; Rance & Young, 2010). In our case we observed low product titers (3–28 ng/mL) in the transiently transfected cells and expect to achieve the titers of 3–15 µg/mL, i.e., industrially relevant productivities, in stably transfected cells after target gene amplification step, according to our previously published data for FSH (Orlova et al., 2019).

Materials and Methods

Generation of expression constructs with different signal peptides for β-chains

Plasmids previously used for transient transfection experiments (Sinegubova et al., 2024) were employed in this study. All genetic constructs were designed according to the following tricistronic scheme: “β-chain-IRES-α-chain-IRESatt-DHFR”, where IRES is an internal ribosome entry site from the encephalomyocarditis virus, wild-type or attenuated (att), and DHFR is a selection marker murine dihydrofolate reductase (Fig. 1A). Step-out PCR was conducted to replace the NSPs of the β-chains. The pairs of long adapter primers, which encoded the heterologous SPs (HSA-SP, Azu-SP, aSP) and the synthetic Kozak sequence, alongside a reverse primer common to all β-chain ORF areas, were used. After sequencing, the PCR products were subcloned into expression vector plasmids via the AbsI-SpeI/NheI endonucleases.

Figure 1 (A) Genetic constructs encoding β-subunits of gonadotropin hormones with variable signal peptides (*SP, sequences are shown) and α-subunits with native signal peptide (αSP). (B) Experimental outline.

Abbreviations: CHO EEF1 UFR and CHO EEF1 DFR, regions flanking the Chinese hamster EEF1A1 gene, containing promoter, intron, terminator, and polyadenylation signal of EEF1A1 gene; *SP, β-chain variable signal peptide; β-chain, β-chain ORF; IRESwt, natural internal ribosome entry site of wild-type EMCV; αSP, signal peptide of the glycoprotein hormone α-chain; α-chain, α-chain ORF; IRESatt, attenuated internal ribosome entry site; DHFR, mouse dihydrofolate reductase ORF; FSH, follicle-stimulating hormone; LH, luteinizing hormone; CG, chorionic gonadotropin; TSH, thyroid-stimulating hormone; HSA, human serum albumin; Azu, azurocidin; aSP, signal peptide of the glycoprotein hormone α-chain; MTX, methotrexate, HT, hypoxanthine-thymidine; ELISA, enzyme-linked immunosorbent assay. N-, H-, and C-regions in SPs sequences are identified using the SignalP algorithm.

Cell culture and stable cell line generation

We used the CHO 4BGD host cell line (genotype BAK1−/− BAX−/− BCL2+ BECN1+ GLUL−/− DHFR−/−) obtained in-house from the CHO S (Thermo Fisher Scientific, Waltham, Massachusetts, USA) using two successive rounds of genome editing and described in (Orlova et al., 2022; Kovnir et al., 2023). Cells were cultured in a CO2 incubator (37 °C, 5% CO2, 155 rpm orbital shaking with amplitude 10 mm) in 125-mL Erlenmeyer flasks (VWR Scientific, Radnor, Pennsylvania, USA) in 30 ml of serum-free ProCHO5 medium (Lonza, Switzerland) supplemented with 8 mM glutamine/alanyl-glutamine, 100 mM hypoxanthine + 16 mM thymidine (HT), all supplements from Paneco, Moscow, Russia. Cell viability and concentration were measured using a Countess automated cell counter (Thermo Fisher Scientific, Waltham, MA, USA) after 0.4% trypan blue (Thermo Fisher Scientific, Waltham, MA, USA) staining.

Ten million cells were electroporated with 50 μg of plasmid DNA using the Neon apparatus and Neon electrotransfection kit (both Thermo Fisher Scientific, Waltham, MA, USA). A single pulse of 1,600 V for 10 ms was applied. Five percent by weight of control plasmid pEGFP-N2 (Addgene#6081-1; Clonentech, Mountain View, California, USA) was added to the target plasmid in each transfection. After 48 h in culture, the cells were centrifuged (300 g, 5 min) and transferred to a selective medium. Selection was performed by culturing the cells in the medium lacking HT, but supplemented with 200 nM MTX (Ebewe, Unterach am Attersee, Austria). The cells were passaged in the selective medium every 3–5 days until viability reached 90%. A higher concentration of MTX (2 μM) was then used to amplify target genes for increased glycoprotein hormones expression. The cells were passaged in the selective medium every 3–5 days until viability reached 90%. Both stably transfected and gene-amplified cell pools were cultured in ProCHO5 medium supplemented with 8 mM glutamine in a simple batch mode for 4 days. Samples of cells and culture supernatant were collected and stored at −20 °C before analysis.

Glycoprotein hormone heterodimer quantification using ELISA

Glycoprotein heterodimeric hormones concentrations in the culture supernatants were determined using a sandwich ELISA, the antibodies used are listed in Table 1. The microplates were coated by capture antibodies at 100 ng per well in 100 mM sodium carbonate buffered solution, incubated overnight at +4 °C. The blocking was performed with 3% bovine serum albumin (BSA) in phosphate-buffered saline (PBS) for 1 h at 37 °C. Fifty microliters of conditioned culture medium were diluted with PBS, added to the wells of the microplates along with 50 μl of anti-α-chain antibodies conjugated with horseradish peroxidase (HRP), dilution 1:20,000. Plates were mixed and incubated for 1 h at 37 °C on a thermostatic shaker. After washing five times with 0.1% PBS-Tween 20 (PBST), 100 μl of 3, 3′, 5, 5′-tetramethylbenzidine (TMB) substrate were added. The reaction was incubated for 10 min at RT, stopped with 100 μl of 5% phosphoric acid, and the optical density at 450 nm was immediately measured using a Feyond-A300 Microplate Reader (Allsheng, Hangzhou, China).

Table 1 Monoclonal antibodies used for ELISA for determination of the heterodimeric hormone concentrations in the culture medium (all produced by XEMA-Medica LLC).

Hormone	Capture antibody (anti-β-chain)	Detection antibody (anti-α-chain, HRP conjugate)	
LH	#XL1	#XF1	
FSH	#XF2	#XF1	
TSH	#XTB1	#XF1	
CG	#XH51	#XF1	

Glycoprotein hormone α-chain quantification using ELISA

Glycoprotein α-chain concentration in the supernatants was determined using a sandwich ELISA, the antibodies toward α-subunit of glycoprotein hormones #K003/1* and a conjugate of monoclonal antibodies against the α-subunit of glycoprotein hormones with horseradish peroxidase #K003 (Diatech LLC, Moscow, Russia) were used. The testing procedure was identical to that employed for heterodimeric hormone quantification.

Glycoprotein hormone heterodimer and free chains quantification using Western blotting

Samples of conditioned media (from 4-day batch culture) were concentrated approximately 30-fold using Vivaspin 500 ultrafiltration devices (Sartorius, Goettingen, Germany) with a 5 kDa MWCO polyethersulfone membranes. Gel samples were normalized based on heterodimeric hormone concentrations determined using ELISA, the same amount of each hormone (150 or 30 ng) was applied to each lane. Samples containing 150 ng of heterodimers were loaded on the gel without pretreatment for analyzing the distribution of the target hormone between the heterodimeric state and the free chain state. Samples containing 30 ng of heterodimers were heated at 95 °C for 10 min before loading for analysis of the total chain content. For the TSH samples stained with anti-β-antibodies the loading was 30 ng in all cases. Separation was performed using sodium dodecyl-sulfate polyacrylamide gel electrophoresis (SDS–PAGE, 12.5% acrylamide in separating gel) at 120 V for 20 min followed by 180 V for 90 min. Semi-dry transfer of proteins was performed on the nitrocellulose membrane (GVS, #1215471) using TE 70 PWR Transfer Unit (GE Lifesciences, Chicago, Illinois, USA) at 60 V for 2 h in a modified Towbin solution containing 5.76 g/L Tris-base, 2.95 g/L glycine, 20% ethanol. In the case of TSH samples, the membrane was additionally heated at 95 °C for 20 min in 0.1% PBST solution to improve the antibodies binding to βTSH. Blocking was performed in 5% BSA-PBST solution at room temperature for 1 h or overnight at 4 °C. Membranes were incubated with primary antibodies in 1% BSA-PBS solution for 1 h at room temperature with rocking, washed twice, and incubated with anti-species antibodies-HRP conjugate when necessary. The antibodies used were as follows: for βFSH and βLH - XF2 and XL1 (ХЕМА) + Goat Anti-Mouse-HRP (ab6789; Abcam, Cambridge, UK), for βTSH – r-mAb to TSH beta (ab8155959; Abcam) + Goat Anti-Rabbit-HRP (ab6721, Abcam), for βCG – К002 (Diatech). The membrane was washed with 0.1% PBST solution three times for 15/5/5 min. Pierce ECL Western reagent (Thermo Fisher Scientific,Waltham, MA, USA) was used for detection. The membrane image was captured using a FUSION Solo 6X chemidocumentation visualization system (Vilber Lourmat, France). Band quantification was performed using the TotalLab TL120 Software v2009 (TotalLab, Newcastle upon Tyne, England), absolute peak volumes method, baseline subtracted.

Transgene copy number analysis using qPCR

Target gene copy number was estimated using quantitative real-time polymerase chain reaction (qPCR). Genomic DNA (gDNA) was isolated from 1.5–2.5 × 106 cells using the ExtractDNA Blood & Cells genomic DNA extraction kit (Eurogen) according to the manufacturer’s instructions. Quantitative PCR was performed using BioMaster HS-qPCR Lo-ROX SYBR 2× mix (Biolabmix, Novosibirsk, Russia) on an Applied Biosystems 7500 Real-Time PCR System (Thermo Fisher Scientific, Waltham, MA, USA). The amplification protocol was as follows: pre-denaturation for 10 min at 95 °C, 40 cycles of amplification (10 s denaturation at 95 °C, 15 s annealing at 58 °C, 15 s elongation at 72 °C). Each sample was analyzed in at least three replicates. The single-copy rab1 gene was taken for normalization. The concentration of the plasmids for calibration was calculated using the Endmemo online calculator (ENDMEMO, 2025). Primers (Table 2) were selected using the Benchling platform (Benchling, 2024) and tested for specificity using the Primer-BLAST tool (Ye et al., 2012). Threshold cycles, PCR efficiency, calibration curves, and copy number calculations were performed using the Applied Biosystems software.

Table 2 Primers for insert copy number analysis using qPCR.

ORF	Forward primer	Reverse primer	
α-chain	CACGCTACAGGAAAACCC	TCTTGGACCTTAGTGGAGTG	
β-chain CG	ATGTGCGCTTCGAGTCCAT	GGCAGAGTGCACATTGACAG	
β-chain LH	GCCTCTTGCTCTTACTTCTAC	AATTGTGGTATTGACTGTTATGC	
β-chain FSH	GCCCAAAATCCAGAAAC	ACAATCAGTGCTGTCGCT	
β-chain TSH	TTACATGTGGGCAAGCGATGT	TGGTGGTGTTGATGGTTAGGC	
Rab1	GGTCACATTGTCGGTGTTTCTG	CAGCCCATGCACTGAAGTATTG	

RNA secondary structure analysis

Secondary structure predictions and free energy calculations for the SP-hormone-encoding mRNAs were performed using the RNAfold algorithm (ViennaRNA Web Services, 2024). The analyzed sequences included the complete pre-translational area of the spliced mRNA, the SP ORF (approximately 60 nucleotides), and 120 nucleotides downstream, encoding the N-terminal part of the corresponding β-chains. The length of the mRNA fragment used for calculations was sufficient for folding the nucleotides of the varied SP area into hairpins and removal of the downstream hairpins.

Statistical analysis

Independent two-tailed Student’s t-tests were performed using the GraphPad Prism software 10.3.1 (GraphPad Software, San Diego, CA, USA), available at GraphPad (2024). One-way ANOVA with a post-hoc Tukey HSD test was performed using the LibreOffice software (LibreOffice, Berlin, Germany), available at LibreOffice, 2024.

Results

Genetic constructs encoding hormone chains with variable signal peptides

In this study, we used the same plasmids used to obtain transiently transfected cell pools (Sinegubova et al., 2024). Briefly, the tricistronic constructs p1.1-Tr2-Gon-BIA, based on the p1.1-Tr2 plasmid vector (Sinegubova, Orlova & Vorobiev, 2023), encoded the β-subunits of the glycoprotein hormones with various SPs in the first cistron, the common α-subunit with the NSP in the second cistron, and the selection marker DHFR in the third cistron, as shown in Fig. 1A.

Strong hairpins in mRNA can cause lower protein expression level in a predictable manner (Eisenhut et al., 2020; Weenink et al., 2018). Using the RNAfold algorithm, we calculated values of free energy of the thermodynamic ensemble for the optimal mRNA secondary structures. Correct 5’-parts of all transcripts were determined as the minimal part of mRNA, which gave all hairpin structures, involving the nucleotides from variable parts, i.e., nucleotides encoding SPs. Centroid secondary structures of the SP-gonadotropin-encoding mRNA variants are shown in Fig. S1. For all four hormones, the most energetically stable structures (low energy) were identified when using Azu-SP (Table 3, in bold).

Table 3 Free energy of the thermodynamic ensemble (kcal/mol) for the SP-hormone mRNA secondary structures calculated using the RNAfold algorithm.

The most energetically stable structures (Azu-SP for all for glycoprotein hormones) are shown in bold.

Hormone name	Signal peptide	
NSP	HSA	Azu	aSP	
FSH	−76.48	−72.76	−91.04	−75.97	
LH	−92.62	−76.77	−94.15	−81.21	
CG	−87.92	−78.55	−96.54	−84.06	
TSH	−71.28	−60.24	−76.50	−65.71	

Development of stably transfected cell pools

To estimate whether the data obtained for transiently transfected cell culture would be relevant for stably transfected cell culture, we obtained 16 corresponding stably transfected cell pools. The experiment outline is shown in Fig. 1B. The 16 plasmids (four SPs for each of the four hormones) were transfected into CHO 4BGD cells using electroporation. The plasmids contained the DHFR selection marker in the expression cassette and therefore could be used for the development of stable cell lines when transfected into the DHFR-deficient (DHFR−/−) CHO 4BGD cells (Orlova et al., 2022; Kovnir et al., 2023). MTX is a selective inhibitor of the DHFR enzyme expressed by an encoded selection marker gene. During selection, only cells containing the DHFR selection marker in their genome can survive in the MTX-supplemented medium. The initial selection was performed with a low concentration of MTX (200 nM in all cases). Specific productivities of cells were increased by the target gene amplification step, performed as culturing of stably transfected cell pools in the presence of 2 μM MTX for 2–3 weeks until the cell viabilities reached 90%. The resulting gene-amplified cell pools were passaged simultaneously at 300,000 cells/mL and cultured in batch mode for 4 days, viable cell densities were determined and culture samples were collected. Cell supernatants were analyzed using ELISA and Western blotting; cell pellets were used for analysis of genome-integrated plasmid copy numbers.

Secretion levels of heterodimeric hormones and free chains estimated using ELISA and Western blotting

Hormone concentrations (heterodimers and total a-chains) in the cell culture supernatants were measured by sandwich ELISA. The cell-specific productivities (Qp) were calculated as the titer of the target protein secreted into the culture medium divided by the cell number (Figs. 2A–2D).

Figure 2 Specific productivity (Qp, pg/cell) for stable CHO cell pools expressing glycoprotein hormones under varying β-chain signal peptide as measured using ELISA for heterodimer (A–D) and α-chain (E–H).

(A, E) FSH, follicle-stimulating hormone; (B, F) LH, luteinizing hormone; (C, G) CG, chorionic gonadotropin; (D, H) TSH, thyroid-stimulating hormone. Abbreviations for signal peptides: NSP, native signal peptide of the corresponding glycoprotein hormone β-chain; HSA, human serum albumin signal peptide; Azu, azurocidin signal peptide; aSP, signal peptide of the glycoprotein hormone α-chain. Data are represented as mean of two technical replicates ± standard deviation (SD). P-values are presented on plot (each heterologous SP vs. NSP), independent two-tailed Student’s t-test. Non-significant P-values are not shown.

For FSH, a small (30%) but statistically significant increase in specific cell productivity by heterodimer was observed only when using the Azu SP (2.7 ± 0.08 pg/cell) compared to the NSP (Fig. 2A).

For LH, replacing the signal peptide with heterologous ones in all three cases resulted in a significant (2–4 times) increase in hormone titers. The specific productivity was 2.5 ± 0.4 pg/cell for HSA-LH, 1.8 ± 0.1 pg/cell for Azu-LH, 1.2 ± 0.1 pg/cell for aSP-LH vs. 0.6 ± 0.1 pg/cell for NSP-LH (Fig. 2B). Previously, we have obtained a polyclonal LH producer cell line using a different approach: co-transfection with two separate plasmids encoding α- and β-chains, both chains with their respective NSP. The productivity of this cell pool was comparable to that obtained in the present study—about 0.3 pg/cell (Orlova et al., 2017) but inferior to all three heterologous SPs. In the case of LH, the use of heterologous SPs was found to be an effective way to increase the specific productivity without complex experiments with pairs of co-transfected plasmids and variable selection pressures for two markers.

For CG, in the case of HSA-CG, we achieved a 60% increase in specific productivity (1.0 ± 0.1 pg/cell compared to 0.6 ± 0.01 pg/cell for NSP-CG). In contrast, in the case of Azu-CG, a 60% drop in productivity was observed (Fig. 2C).

In the case of TSH, the productivity of cells in the heterodimeric form of the hormone increased by 7–13 times when using all three heterologous signal peptides, with the maximum productivity of 1.6 ± 0.05 pg/cell for HSA-TSH (Fig. 2D).

As previously described (Sinegubova et al., 2024), in transiently transfected cultures, the relative levels of secreted α-subunit decreased when the NSPs of the β-subunit were replaced with heterologous ones. For the stable cell pools, we also measured α-chain concentrations using ELISA (Figs. 2E–2H). This analysis allowed us to accurately measure the total α-chain concentration, i.e., the sum of the β-chain-bound α-chain and the free α-chain. Since the signal peptide of the α-chain was the same in all plasmids studied, we expected the same level of total α-chain in all 16 cases. In fact, the α-chain level varied with changes in the SPs for the β-chains. Statistically significant differences in total α-chain were observed for LH (Azu vs. NSP, Fig. 2F) and TSH (Azu vs. NSP, Fig. 2H).

Therefore, for the stably transfected pools, we performed Western blotting to further investigate the distribution of the α- and β-chains between the freely secreted form and the heterodimeric form. We studied the samples in two ways: without any pretreatment to see the distribution of the chains secreted as the αβ-heterodimer and the freely secreted form, and after heating samples in sample buffer at 95 °C for 10 min—as fully dissociated chains. Blotting bands intensities of the total heat-dissociated chains were considered as the total levels of corresponding chains in the culture medium. The relative intensities of the free chains bands in the non-treated samples were considered as the level of free chain secretion. Medium samples were not treated with DTT because all reduced chains were not recognized by the antibodies used (data not shown). According to the control lanes with the purified hormones, no heterodimer dissociation was visible without heat treatment and no heterodimers remained after heat treatment in all cases.

For FSH, the increase in titer for Azu-FSH corresponded to a decrease in the free β-chain secretion level. This level was reduced almost twofold (from 32% to 13%, Figs. 3A, 3I). The secretion level of the free α-chain remained less than 2% in all cases. For FSH, CG, and TSH, very small amounts (1–9% of the total) of free α-chains were observed in the culture medium (Figs. 3A, 3C, 3D) in all cases except for the NSP-TSH cell pool, which secreted 66% free α-chain. In contrast, for LH, significant amounts (22–64%) of free α-chain were observed for all four signal peptides (Figs. 3B, 3J). In the case of CG (Fig. 3G), the free β-chain level remained relatively constant (36–41%) when varying their signal peptides. It is noteworthy that LH and CG possess nearly identical β-subunits, with CG containing an additional C-terminal domain with O-glycosylation sites, called CTP. This study suggests that this additional domain significantly alters the behavior of the pre-β-chain within cellular compartments prior to secretion. The β-chain of LH appears to be unstable, leading to a large amount of the pre-α-chain remaining in the free state in cellular secretory pathways. In the case of NSP-TSH, Western blotting data (Fig. 3D) and ELISA results (Fig. 2D) revealed unusually low levels of secreted TSH heterodimer. Simultaneously, a significant excess () of free α-chain was observed (66%, Figs. 3D, 3I). The excess was significantly reduced to 6–10% when heterologous SPs were used, coinciding with low levels of free β-chains (1–3%, Figs. 3H, 3I) and a substantial increase in heterodimeric hormone titer (ELISA, Fig. 2D).

Figure 3 Western blot analysis of conditioned media from stable cell pools expressing glycoprotein hormones under varying signal peptides of the β-chains.

Antibodies against α-chain (A–D) and β-chains (E–H) of the corresponding hormones. (A, E) FSH, follicle-stimulating hormone; (B, F) LH, luteinizing hormone; (C, G) CG, chorionic gonadotropin; (D, H) TSH, thyroid-stimulating hormone. Abbreviations: NSP, native signal peptide of the corresponding glycoprotein hormone β-chain; HSA, human serum albumin signal peptide; Azu, azurocidin signal peptide; aSP, signal peptide of the glycoprotein hormone α-chain; h/d, heterodimer. Loading: 150 ng/well without pretreatment (−), 30 ng/well after denaturation at 95 °C for 10 min (+); for β-TSH 30 ng in all cases. Arrows indicate positions of h/d and free chains. Standards used: follitropin alfa (recombinant hFSH), lutropin alfa (recombinant hLH), choriogonadotropin alfa (recombinant hCG). (I–L) Band percent intensity for α- and β-chains expression (free chain to total chain) as quantified using TotalLab Software.

We hypothesize that the NSPs of the TSH and LH β-chains are inefficient, and this “ineffective” biosynthesis of their β-chains may contribute to the fine regulation of TSH and LH production in the anterior pituitary.

Copy number of integrated into genome plasmids estimated using qPCR

To investigate whether increased hormone titers were due to higher copy numbers of genome-integrated plasmids, we performed quantitative PCR (qPCR) analysis of cellular DNA (Fig. 4). For FSH and LH, the integrated plasmid copy number ranged from 10 to 33 copies per haploid genome when measured by the β-chain ORF. No significant differences were observed between various plasmid variants when comparing copy numbers for both β- and α-chains. However, in the case of NSP-CG, the copy number of both β and α-chains was significantly higher than other plasmids, which aligns with the enhanced level of secreted β-chain observed by Western blotting.

Figure 4 Copy numbers of α-chain (left, no fill) and β-chain (right, solid fill) ORFs integrated into the producer genome.

FSH, follicle-stimulating hormone; LH, luteinizing hormone; CG, chorionic gonadotropin; TSH, thyroid-stimulating hormone. Normalization on the single-copy rab1 gene. Data are represented as mean of four replicates ± SD. The lowercase letters “abcd” and “efgh” represent number of independent groups according to one-way ANOVA with post-hoc Tukey HSD test. P-values are presented on plot.

Despite the increased copy number of β-chain ORF copies, no corresponding increase in β-chain synthesis or heterodimer production was observed. This discrepancy may be attributed to genetic cassette fragmentation during genomic amplification, potentially leading to transcriptionally inactive regions within the integrated plasmid. Furthermore, the copy numbers for the two pairs of plasmids exhibiting the highest protein titers, NSP-LH–HSA-LH and NSP-TSH–HSA-TSH, were nearly identical. This finding suggests that increased copy number alone cannot fully explain the significant increase in hormone heterodimer secretion levels.

Discussion

In the classical secretory pathway, translocation of secreted proteins into the ER lumen can be the main rate-limiting step. This bottleneck can be overcome by changing the SP (Zhang, Leng & Mixson, 2005). According to several studies, a single universally translocation-effective SP does not exist for proteins of different classes of the same or different biological species (Kapp et al., 2013). We have shown that within the same protein family—glycoprotein hormones, where the α-chain is identical and the β-chain has some homology between four family members—the same heterologous SP for its β-chain has a distinct or even an opposite effect on specific productivities in the CHO cell culture. For example, using the HSA SP resulted in a 60% titer increase for CG and a 13-fold titer increase for TSH, and the Azu SP increased secretion for all glycoprotein hormones except CG. A similar phenomenon was observed by Haryadi et al. (2015) who found that the strength of one heterologous SP differed for light chains of monoclonal antibodies with similar amino acid composition and physicochemical properties. It is obvious that for the effective heterodimeric protein production, not only high absolute expression levels of the individual subunits are important, but also their optimal ratio, i.e., certain stoichiometry (Pybus et al., 2014; Mathias et al., 2020).

The assembly of glycoprotein hormone heterodimers occurs through different mechanisms, with significant variations between CG and LH, which have highly homologous β-subunits. In the case of CG, FSH, and TSH, independent folding of α- and β-subunits occurs, after which the heterodimer is assembled in the ER using the “threading” mechanism (Xing et al., 2004; Bernard et al., 2014). In the case of LH, the α-chain folds independently, and the β-chain folds only in the presence of the α-chain according to the “wraparound” mechanism, i.e., the α-chain acts as a chaperone for the β-chain (Bernard et al., 2014). In our study, a significant accumulation of free α-chain was visible by blotting in the case of LH (Figs. 3B, 3J). This phenomenon can be explained by the limited resources of CHO cells, in contrast to pituitary cells, for heterodimer assembly. The independently folded α-chain is secreted into the culture medium in large quantities, while the β-chain, regardless of the SP used, does not fold efficiently in the absence of the α-chain excess in the ER and fails to assemble into the heterodimer. Therefore, our study confirms that optimal heterodimer expression is a complex function of the cell’s protein synthesis and assembly capabilities.

For the human glycoprotein hormone family, we identified an optimal SP - from human serum albumin, which led to an increase in productivity for three out of four glycoprotein hormones and was not inferior to the NSP of the FSH β-chain. This effect was observed in both the transiently and stably transfected CHO cell pools, although the magnitude of the productivity change for transient and gene-amplified cell pools was not the same. In the case of LH and TSH, we observed high levels of free α-chain in the culture medium when the β-chain was expressed with its NSP. These data suggest that the rate-limiting step in LH and TSH heterodimer assembly is the concentration of the β-chains in the ER lumen.

In the work of Kober, Zehe & Bode (2013), the productivity of the CHO K1 cell lines was increased by 50–100% by replacing the NSPs of the light and heavy chains of the immunoglobulin G with the SPs of HSA and Azu, which is in line with our observations about glycoprotein hormones.

We assume that integration of expression constructs into the genome occurred at random sites for all stably transfected pools. The average amplification efficiency and transcription level for the obtained polyclonal lines should not differ significantly because the genetic constructs for each hormone differs only in the SPs of the β-subunits ORFs. According to the ELISA, Western blotting, and qPCR results, differences in the hormone heterodimeric forms secretion level cannot be attributed to increased α-chain synthesis or by differences in the copy number of genetic constructs integrated into the genome. Therefore, our approach demonstrates that the increaseв specific cell productivity for hormone heterodimers is driven by enhanced translation and translocation efficiency of the β-subunit into the ER by coupling them with heterologous SPs.

It is known that even small variations in the gene structure may have a significant impact on the final protein expression level which is due to the presence of some hairpins interfering with ribosome binding, translation initiation complex assembly, and elongation. The calculated values of the mRNA free energies varied for all 16 SP-hormone combinations, but these changes hardly correlated with the differences in hormone titer. For all four hormones, the most stable structure of the mRNA thermodynamic ensemble was observed when using the Azu SP, but only the Azu-CG stable cell pool demonstrated the decrease in titer compared to NSP. In contrast, for the Azu-FSH, Azu-LH, and Azu-TSH stably transfected cells we observed a 1.3 to 8.2-fold increase in heterodimer titers.

For all glycoprotein hormones, the results obtained for the transient transfectants moderately correlated with the results for the stably transfected cell pools. The DHFR/MTX-mediated selection is a standard method for gene amplification in stably transfected CHO cells. Addition of the DHFR inhibitor MTX enables selection of the cells with the increased levels of the DHFR selection marker, leading to correspondingly higher expression of target proteins. During genomic amplification, an increase in the number of integrated into the genome plasmids occurs. In our study, we observed over a 100-fold increase in the titer for each hormone secreted by the stable cell pools (selection pressure 2 μM MTX) compared to the transiently expressing cells. Such a strong difference in the hormone heterodimer secretion for the stably transfected pools can be explained by a significant increase in the load on the cell protein synthesis machinery. Before obtaining stably transfected cell lines, it is possible to preliminarily assess the impact of the chosen SP on the target protein secretion level in transiently transfected cell culture. This approach can significantly reduce experimental time and effort required.

Conclusions

For stably transfected CHO cell pools expressing four glycoprotein hormones, replacing the native β-chain signal peptide with a heterologous one resulted in increased cell productivity. For CG, LH and TSH, a 1.5-, 4- and 13-fold increase in cell productivity, respectively, was observed when using the HSA signal peptide. For FSH, a 60% increase in productivity was observed when using the Azu signal peptide. qPCR and Western blotting results demonstrated that the observed increase in productivity was not due to an increased number of plasmid copies integrated into the genome, a general increase in the biosynthesis levels of both chains or removal of strong hairpins in the mRNA. Increasing the biosynthesis level of β-chains glycoprotein hormones by replacing natural signal peptides with heterologous ones facilitates the development of producer cell lines with significantly increased specific productivity.

Supplemental Information

Supplemental Information 1 Centroid secondary structures of the SP-gonadotropin-encoding mRNA variants predicted using the RNAfold algorithm.

The analyzed sequences included the complete pre-translational area of the spliced mRNA, the SP ORF (approximately 60 nucleotides), and 120 nucleotides downstream, encoding the N-terminal part of the corresponding β-chains. The structures are colored by base-pairing probabilities from blue (zero probability) to red (100% probability). The SP position (first codon) is indicated with a red arrow.

Supplemental Information 2 Raw ELISA data for the determination of heterodimers and alfa-chain concentracion and p-values.

Supplemental Information 3 Raw Western blot images.

Supplemental Information 4 Calculation of free chain content visible by blotting using TotalLab.

Supplemental Information 5 Raw qPCR data and ANOVA p-values.

We thank Yaroslava Kochina for obtaining the polyclonal pools producing TSH and CG. We thank Lutsiya Dayanova for assistance in the molecular cloning. ChatGPT 3.5 was used for the English language editing.

Additional Information and Declarations

Competing Interests

The authors declare that they have no competing interests.

Author Contributions

Maria V. Sinegubova performed the experiments, analyzed the data, prepared figures and/or tables, authored or reviewed drafts of the article, and approved the final draft.

Denis E. Kolesov performed the experiments, analyzed the data, authored or reviewed drafts of the article, and approved the final draft.

Ivan I. Vorobiev conceived and designed the experiments, authored or reviewed drafts of the article, and approved the final draft.

Nadezhda A. Orlova conceived and designed the experiments, prepared figures and/or tables, authored or reviewed drafts of the article, and approved the final draft.

Data Availability

The following information was supplied regarding data availability:

Raw ELISA and qPCR data, as well as the original Western blot images, are available in the Supplemental Files.

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
