# Peer review of "Increased glycoprotein hormone yield in stably transfected CHO cells using human serum albumin signal peptide for beta-chains"

_PeerJ, doi:10.7717/peerj.18908_

## Round 0.1 · original submission · Major Revisions

In response to your appeal, we would request you to kindly address the reviewer's comments and re-submit the manuscript.

· Appeal

Appeal


· · Academic Editor

Reject

As part of the review process, it has been brought to my attention that the authors have recently published a very similar work (https://link.springer.com/article/10.1134/S1607672923700576). We aren't able to consider this submission because it isn't self-contained and doesn't represent a minimum publishable unit.

Reviewer 1 ·

Basic reporting

The article by Sinegubova et al focusses on how glycoprotein hormone family productivity can be increased in CHO cells. The article is insightful as it finds better ways to increase cell specificity of these hormones.
Minor comments:
1. The article is largely well written but needs minor English improvements. Line 51 - needs to be changed to widely used. Line 81- "Probably in the near future they will find application in" who is they here? Such sentences should be largely avoided.
2. Lines 84 - 86 needs a reference
3. Figure 2 -The lines indicating significance between bars are crossing through the bars. This is a very unaesthetic way of showing graphs and it would be helpful if the authors move the significance lines above the bars like they have done in some panels.
4. For figures 2-4, it is vital that the authors have labelling for each graph. Eg: in figure 2, panel A includes four graphs. This gets very confusing in the results, it is helpful if each graph was given an alphabet, so you would have panels A-H instead of just A and B in figure 2.

Experimental design

In general the overall experimental designs described in the paper seem sound. However more clarifications are needed to help make the methods section more rigorous.

Minor comments:
1. Methods line 111 - Although the authors provide a reference for the construct used, it would be needed that the authors also give a brief overview of this contract so that the readers do not have to rely on downloading another paper to understand design of this paper.
2. For the Western blots- What was the loading control used? Please specify. Moreover, what technique was used to quantify protein content (such as BCA?). Please mention this and share data from this as well.
3. For the statistical analysis- You need to provide exact p - values in the respective figures (this is in the figure legends where you report statistical significance) and not in a supplemental file. Also please mention the version of GraphPad used for analysis.
4. For figure 2- Please mention the kind of statistical testing done.

Validity of the findings

There are some major comments that the authors need to address for the paper to be accepted.

Major comments:
1. Figure 3- This is a western blot figure with only blots being shown and data interpretation seems to have been done based on what is seen from the eye (lines 223-229). This is definitely not an acceptable format for western blotting. The authors would need to quantify these results using appropriate softwares (imageJ, imageStudi Lite etc) and show the blot along with the quantified graphs. This will also help remove words such as "low levels" and "significant amounts" in lines 223-229 as such words cannot be used without proper statistical testing and data quantification.

Additional comments

The figure legends need to be more comprehensive, things like p values, statistical testing, colors for each group etc need to be mentioned.

Reviewer 2 ·

Basic reporting

The present paper by MV Sinegubova et al. deals with the optimization of human recombinant glycoprotein hormones production and secretion through the replacement of their original β-subunit signal peptide (SP) sequences by alternative ones (particularly the human serum-albumin SP sequence).

General comment
The aim of the present paper is the same as a previous one by these authors except it is performed in stable expressing CHO cells instead of transiently-expressing CHO cells. For this reason, the methodology for genomic insertion of the plasmids into the CHO cells’ genome should be more clearly described and discussed.

Specific comments
lines 65-66 : the reference 20 appears attractive. Has this technique been used for the GPH subunits’ aa sequences ? If yes, what were the results ?
lines 103-104 : it is interesting to indicate the fold times increases of productivity but also the absolute values. It could be expected that it is easier to increase production when it is naturally low. Is that the case ?
line 125 : precise what the use methotrexate is for and how the increase in its concentration works for the present use.
Figure 3B CG : The hCG alfa beta-subunit is not detected upon heat dissociation. Does th antibody used recognize both the alpha-associated and free beta ?

Conclusions
This work is well-conducted and its presentation is fine in terms of language, illustrations and references.
Nevertheless, the conclusions are not very different from the previous paper from the authors (ref 28). and should more insist on the interest of stably-expressing cells, and how they could be improved, for commercial production.

Experimental design

The experimental design is fine.

As mentionned above, the difference with previous work from these authors (ref 28)nis that the cells stably express the glycoprotein hormones.

The methods for getting interesting stably expressing cells should be highlighted and described more precisely.

Validity of the findings

The study of signal peptide sequences for improving recombinant glycoproteins' expression should be more broadly apprehended (on the basis of ref 20 for example).

---

## Round 0.2 · Major Revisions

Thank you for re-submitting the manuscript addressing all the reviewer's comments. However, as agreed with one of the reviewers, I would request the authors to replace or re-do the Western Blot in the figure 3H (the figure number as per the original version). Overall, the quality of the Western Blot should be improved otherwise it will impact the final analysis. I strongly recommend to either re-do the Western Blot or replace the figure if you have a better representative one.

Reviewer 1 ·

Basic reporting

This paper by MV Sinegubova et al. reports optimization of human recombinant glycoprotein hormones production and secretion through the replacement of their original β-subunit signal peptide (SP) sequences by alternative ones. Overall the paper is well written and the authors have answered my concerns on the previous version as well as taken some of the things I suggested into consideration. The paper definitely has improved in quality and is getting ready for publication. However, I still do have some concerns on the western blots. Please find them in the subsequent sections.

Experimental design

I am a bit worried about the western blot shown in Figure 4 panel H. The blot show here is indicating the B- chain. However, the lane in which the B-chain is pointed towards, the band for 30ng looks very pixelated. In fact to my eye it does not look like a complete band. I am unsure if this is a representative image, but I am wondering if the authors are able to re-run the western and see if the band is actually truly visible and maybe analyze the new blot.
With the current blot I am concerned that since the band does not look complete it may be leading to an error in analysis?

Validity of the findings

Please see my above comment. But in general I would suggest to re-validate the western blot run and analysis for Figure 4 panel H.

Additional comments

None

Reviewer 2 ·

Basic reporting

The authors have adequately taken my comments in consideration in this second version.

Experimental design

The authors have adequately taken my comments in consideration in this second version.

Validity of the findings

The authors have adequately taken my comments in consideration in this second version.

---

## Round 0.3 · accepted · Accept

Thank you for resubmitting the revised manuscript addressing all the reviewer's comments.

Reviewer 1 ·

Basic reporting

no comments

Experimental design

no comment

Validity of the findings

no comment